

# Modeling inter-continental transport of ozone in North America with CAMx for the Air Quality Model Evaluation International Initiative (AQMEII) Phase 3

Uarporn Nopmongcol[1], Zhen Liu[1], Till Stoeckenius[1], Greg Yarwood[1]

[1]Ramboll Environ, 773 San Marin Dr., Suite 2115, Novato, CA 94945

*Correspondence to*: Dr. Uarporn Nopmongcol (unopmongcol@ramboll.com)

**Abstract.** Inter-continental ozone ($O_3$) transport extends the geographic range of $O_3$ air pollution impacts and makes local air pollution management more difficult. Phase 3 of the Air Quality Modeling Evaluation International Initiative (AQMEII-3) is examining the contribution of inter-continental transport to regional air quality by applying regional scale atmospheric
models jointly with global models. We investigate methods for tracing $O_3$ from global models within regional models. The CAMx photochemical grid model was used to track contributions from boundary condition (BC) $O_3$ over a North America modeling domain for calendar year 2010 using a built-in tracer module called RTCMC. RTCMC can track BC contributions using chemically reactive tracers and also using inert tracers in which deposition is the only sink for $O_3$. Lack of $O_3$ destruction chemistry in the inert tracer approach leads to over estimation biases that can exceed 10 ppb. The flexibility of
RTCMC also allows tracking $O_3$ contributions made by groups of vertical BC layers. The largest BC contributions to seasonal average daily maximum 8-hour averages (MDA8) of $O_3$ over the US are found to be from the mid-troposphere with small contributions from the upper troposphere-lower stratosphere. Contributions from the lower troposphere are shown to not penetrate very far inland. Higher contributions in the Western than the Eastern US, reaching an average of 57 ppb in Denver for the 30 days with highest MDA8 $O_3$ in 2010, present a significant challenge to air quality management approaches
based solely on local or US-wide emission reductions. The substantial BC contribution to MDA8 $O_3$ in the Intermountain West means regional models are particularly sensitive to any biases and errors in the BCs. A sensitivity simulation with reduced BC $O_3$ in response to 20% lower emissions in Asia found a near linear relationship between the BC $O_3$ changes and surface $O_3$ changes in the Western US in all seasons and across the US in fall and winter. However, the surface $O_3$ decreases are small: below 1 ppb in spring and below 0.5 ppb in other seasons.

## 1 Introduction

Inter-continental ozone ($O_3$) transport extends the geographic range of $O_3$ air pollution impacts and makes local air pollution management more difficult (Jaffe et al., 2003; Zhang et al., 2011; Emery et al., 2012). The Air Quality Model Evaluation International Initiative (AQMEII) aims to better understand uncertainties in regional-scale model predictions and foster continued model improvement by providing a collaborative, cross-border forum for model development and evaluation in



North America and Europe (Galmarini and Rao, 2011). While phases 1 and 2 of the AQMEII focused on performance of different types of regional-scale models, Phase 3 (AQMEII-3) examines the contribution of inter-continental transport to regional air quality by applying regional scale atmospheric models jointly with global models (Galmarini et al., 2017). Other AQMEII-3 objectives include assessing the sensitivity of regional transport to emissions changes in key source regions

world-wide and inter-comparing the performance of global and regional-scale models. Multiple models were applied in the AQMEII-3 for North American (NA) and Europe (EU) regional domains with each model required to track the inflow of $O_3$ from the lateral domain boundaries at various vertical heights. As in previous Phases of AQMEII the organizers made available key model input data to promote consistent model applications and simplify interpretation of results across participating models.

We applied the Comprehensive Air Quality Model with Extensions (CAMx) photochemical grid model (Ramboll Environ, 2015) for the NA domain using model inputs provided by the AQMEII. Our simulations address the first two objectives of AQMEII-3 by evaluating contributions introduced via boundary conditions (BC) as inert tracer (i.e., excluding photochemical removal process) and examining $O_3$ response to reduction of anthropogenic emissions in East Asia or globally. Our study is unique in that we are the only AQMEII-3 participants to also track BC $O_3$ using chemically reactive

tracers.

## 2 Methodology

### 2.1 Base Case Modeling

Air quality modeling for the NA domain and calendar year 2010 used CAMx version 6.2 (Ramboll Environ, 2015) to simulate formation and transport of $O_3$. Gas-phase chemistry was included using the Carbon Bond (CB05) mechanism

(Yarwood et al., 2005) but heterogeneous-phase chemistry was not included for efficiency and because the focus is on $O_3$. The CAMx modeling domain covers the Continental US with 459 by 299 grid cells of 12 by 12 km resolution and 26 vertical layers (Table S1). The vertical height of the first layer is 20 meters. Model inputs were prepared from data provided to all AQMEII participants supplemented by other data sources (described later). The 2010 annual simulation was initialized on December 22, 2009, to limit the influence of initial concentrations.

This study uses reactive tracers with a chemical mechanism compiler (RTCMC) in CAMx to track contributions from BCs. The RTCMC module simulates explicit tracers in parallel to the host model and represents sources (emissions and BCs), transport processes (advection and diffusion), deposition and user-defined chemistry (Yarwood et al., 2014). RTCMC chemistry can use species concentrations from the host model, e.g., OH radical, in the reactions of tracers. If no chemistry is

defined for an $O_3$ tracer, then it becomes chemically inert and deposition is the only sink. Currently, RTCMC models dry deposition of tracers but not wet deposition which will result in conservative estimates of BC contributions.





CAMx can also track BC $O_3$ contributions through the ozone source apportionment technology (OSAT; Ramboll Environ, 2015) module but we use RTRAC because it offers flexibility to model reactive and inert tracers, and to track BC $O_3$ from specific groups of vertical layers. Vertical attribution is valuable in identifying height ranges of transported $O_3$ that are most influential to ground-level $O_3$. Baker et al. (2015) compared RTCMC and OSAT over North America and found that the two

approaches estimated very similar BC $O_3$ contributions in warmer months. Baker et al. also evaluated computational efficiency of RTCMC tracers. We use the same RTCMC scheme for reactive $O_3$ tracers as Baker et al. (Table S2) which includes $O_3$ destruction by odd-hydrogen (i.e., $HO_2$ and OH) and alkenes but not $O_3$ destruction by NO because this would entail tracking conversion of BC $O_3$ to/from several NOy species including $NO_2$, PAN and $HNO_3$. OSAT accounts for $O_3$ destruction by NO and Baker et al. show that omission of $O_3$ destruction by NO in the RTCMC scheme causes some positive

bias in BC $O_3$ estimates in winter months (Baker et al., 2015).

The BC $O_3$ contributions are analysed in terms of MDA8 by season because the MDA8 is relevant to the US National Ambient Air Quality Standard (NAAQS) for $O_3$ which is set at a level of 70 ppb. Seasonal averages are used to evaluate how BC contributions depend upon transport patterns and photochemistry.

**2.1.1 Meteorology**

Meteorological data for calendar year 2010 were developed by the US Environmental Protection Agency (EPA) for AQMEII phase 2 using the Weather Research Forecast (WRF; Skamarock et al., 2008) model with 12 km resolution. The WRF domain was defined in Lambert Conformal Projection with 471 by 311 grid cells and 35 vertical layers with a 20 meter deep surface layer. The WRF physics options were described in Gilliam et al., 2012. The WRFCAMx pre-processor reformatted

WRF output for CAMx and diagnosed vertical mixing parameters. CAMx employed fewer vertical layers (26) than WRF (35) to reduce the computational burden of the air quality simulations. The CAMx vertical layers exactly matched those used in WRF for the lowest 10 layers (up to 577 m); above this height several WRF layers were combined to single CAMx layers (Table S1). The minimum vertical diffusivity ($K_v$) was set to 0.1-1.0 $m^2/s$ based on input landuse.

**2.1.2 Emissions**

Anthropogenic and fire emissions for 2010 were provided by AQMEII (Pouliot et al., 2015) separately for surface emissions and six elevated source groups, namely fires, international marine shipping, electric generating units (EGU), other point sources (non-EGU), Mexico point sources, and Canada point sources. Biogenic emissions were obtained from the Model of Emissions of Gases and Aerosols from Nature version 2.1 (MEGAN; Guenther, et al., 2006; Sakulyanontvittaya, et al., 2008). MEGAN has a global database of landcover derived from satellite data at 1 km resolution. Meteorological input data

for MEGAN (i.e., temperature and solar radiation) were taken from the WRF predictions. Annual emissions in 2010 in the modeling domain for each source sector are summarized in Table 1.





### 2.1.3 Boundary Conditions

BCs were provided by the European Centre for Medium-Range Weather Forecasts (ECMWF). The ECMWF BC data were based on the Composition-Integrated Forecast System (C-IFS) model (Flemming et al., 2015) which output three-hourly, three-dimensional gridded concentrations which were formatted for CAMx.

We tracked contributions from $O_3$ BCs in three height ranges defined at the CAMx boundaries, namely from layers 1 to 16 (layers below 750 mb; lower troposphere; LT), 17 to 23 (layers between 750 mb and 240 mb; middle troposphere; MT), and 24 to 26 (layers above 250 mb; upper troposphere and lower stratosphere; UTLS). We introduced both reactive and inert $O_3$ tracers for the same vertical groups. Reactive tracers undergo chemical decay according to chemistry scheme defined using

the RTCMC, whereas inert tracers only participate in physical processes with no chemical removal. Table S2 presents the RTCMC chemistry scheme for reactive tracers as well as the physical properties for both reactive and inert tracers.

### 2.2 Sensitivity Scenarios

Two sensitivity simulations defined by AQMEII were conducted to quantify $O_3$ response to anthropogenic emission reductions. The GLO scenario reduced all anthropogenic emissions by 20% globally and the EAS scenario reduced

anthropogenic emissions in the East Asia by 20%. In the GLO scenario, the CAMx NA anthropogenic emissions were reduced by 20% for the entire modeling domain with no changes to fire and biogenic emissions. Under the EAS scenario the CAMx NA emissions are the same as in the base case. ECMWF provided BCs specific for each scenario. The ECMWF BCs inadvertently omitted $H_2O_2$ for the EAS scenario and so we followed AQMEII suggestion to use $H_2O_2$ from the base case for both the EAS and GLO scenarios. Other modeling inputs are unchanged from the base case. The annual CAMx sensitivity

simulations were conducted with the inert and active boundary $O_3$ tracers described above. Several metrics are used when comparing sensitivity scenarios to the base case including the 4[th] highest maximum daily 8-hour average (H4MDA8), the average of the thirty highest MDA8 $O_3$ days (Top 30), and seasonal average MDA8. We show the results for the entire modeling domain and additionally discuss our findings for 22 selected major cities in a wide variety of climatic and geographic environments (Figure S1). Cities are represented by the monitoring site with highest H4MDA8 in the

metropolitan statistical area.

### 3 Results

### 3.1 Model Performance Evaluation (MPE) of Ozone

We evaluate $O_3$ model performance for the base case relative to benchmarks that are accepted in NA. Predictions of MDA8 $O_3$ are evaluated against observations from the Clean Air Status and Trends Network (CASTNET; rural) and the Air Quality

System (AQS; urban and rural). Observations are considered valid when at least 75 percent of data are available. A 40 ppb





$O_3$ cutoff is applied to focus on the upper end of $O_3$ frequency distributions. Statistical metrics used in this evaluation include two bias metrics, normalized mean bias (NMB) and fractional bias (FB); and two error metrics, normalized mean error (NME) and fractional error (FE) (Table 2). Table 3 shows that model biases (NMB, FB) are less than ±12% and errors (NME, FE) up to 15% for all seasons and both networks, well within the bias/error goals of less than ±15% and 35%

recommended by EPA (1991). The model tends to overestimate $O_3$ in spring and summer, while underestimate in fall and winter. Slightly better performance is found at CASTNET sites than at AQS sites, which include many urban sites that are heavily impacted by local emissions not well resolved by our modeling grid.

The seasonal spatial distributions of NMB, NME, and Pearson's correlation coefficient (r) at individual CASTNET and AQS

sites are shown in Figures S2 – S5. The springtime performance is weakest ($|r| < 0.5$) in the Western US where high terrain prevails. Stratospheric $O_3$ intrusions are influential in the high terrain of the Western US during spring (Langford et al., 2009; 2015; Lin et al., 2012b; Emery et al., 2012) but models do not always represent this process accurately.  In summer, the performance is weakest along the Gulf coast and California coast. Model performance against AQS observations is presented separately for selected 22 major cities. CAMx performs well at these cities, with NMB of -12% to 12% and NME

of 12% to 21% for MDA8 with zero threshold (Table S3), satisfying the bias/error goals. The performance statistics improve when applying a 40 ppb threshold (Table S4).

We evaluated vertical profiles of $O_3$ at Trinidad Head (41.0541°, -124.151°) on the northern coast of California (Figure S6-S7). This location is indicative of $O_3$ entering the US with prevailing winds from the Pacific Ocean and useful for evaluating

$O_3$ BCs over the Pacific. CAMx reproduces the strong gradient in $O_3$ starting at the tropopause (between ~10 and ~14 km) but sometimes underestimates $O_3$ near the model top (~ 16 km). In the free troposphere (~1 to ~10 km), CAMx performance is reasonable with a mix of over and under-predictions. In the marine boundary layer (MBL; < 500 m), CAMx tends to over predict surface $O_3$ because the model lacks a consistently observed gradient toward lower $O_3$ at the surface.  Potential causes are too little $O_3$ deposition to the ocean, lack of $O_3$ destruction by halogen chemistry in the MBL, too strong vertical mixing

in the MBL, or a combination of factors. The influence of low bias in the Pacific MBL will be confined to West Coast States because this air is blocked effectively by western mountain ranges (see Section 3.5). However, similar biases in the MBL over the Gulf of Mexico and Atlantic Ocean could influence $O_3$ across the Southern and Eastern US (see Section 3.2). Further analysis is required to address this uncertainty.

### 3.2 Regional MPE Analysis at Remote Sites along the Gulf Coast

We found a consistent high bias for summer $O_3$ at sites near the Gulf coast and investigated potential causes using our tracer simulation results. Global modeling studies have reported high $O_3$ bias simulated in air arriving along the Texas coast (Fiore et al., 2002; 2014; McDonald-Buller et al., 2011; Zhang et al., 2011). The high $O_3$ bias has been reported by a regional modeling study (Sarwar et al., 2015; Smith et al., 2015). Adding reactions of halogens (especially iodine; Smith et al., 2015)





to the $O_3$ chemistry in CAMx mitigated but did not eliminate this bias. This study provides an opportunity to examine whether bias in $O_3$ BCs could contribute to $O_3$ bias along the Gulf coast.

Time series of daily MDA8 $O_3$ together with $O_3$ BC contributions are shown in Figure 1 for Galveston and Sabine Pass in Texas. CAMx tends to over predict $O_3$ during summer months when onshore winds are prevalent (TCEQ, 2016) due to the enhancement of Bermuda High bringing warm air from the Gulf of Mexico (Zhu and Liang, 2013). During this period CAMx has a large over prediction bias exceeding 15 ppb at the two sites when observations are low (~20 ppb). Reactive $O_3$ tracers exceed observed $O_3$ by an average of 2-3 ppb which explains only a portion of the total model bias so other factors must be contributing (not examined here). Inert $O_3$ tracers are on average higher than the reactive tracers by 10-12 ppb, demonstrating that inert tracers can over estimate BC contributions.

### 3.3 BC Ozone Contributions to Surface Ozone

BC contributions based on the active tracers to seasonal average MDA8 $O_3$ over the US are higher in spring than summer (Figure 2). Spring contributions are over 40 ppb across the Western US which is the region most influenced by pollution transported from Asia (Jaffe et al., 2003; Zhang et al., 2011; Lin et al., 2012; Emery et al., 2012). In particular, high $O_3$ events over the high terrain in the Western US have been linked to intercontinental transport and stratospheric intrusions (Lin et al., 2012a, b). The highest modeled contributions occur in spring which is consistent with observations (Parish et al., 2012; Cooper et al., 2012) and previous modeling studies (Emery et al., 2012; Fiore et al., 2014). BC contributions in the Eastern US are generally below 40 ppb in spring and below 30 ppb in other seasons.

BC contributions on high $O_3$ days are compared to the summer average BC contributions in Figure 3 for 22 major cities. The metrics for high $O_3$ are the H4MDA8, which is relevant to the NAAQS but is a single day, and the Top 30 which includes a variety of conditions that can lead to high $O_3$. They are compared to the summer average MDA8 $O_3$ metric which includes high and low $O_3$ days. In 15 of the 22 major cities (Los Angeles, Sacramento, Dallas, Kansas City, St. Louis, Chicago, Atlanta, Cincinnati, Columbus, Detroit, Pittsburgh, Baltimore, Philadelphia, New York, and Boston), the BC contribution to the Top 30 and summer average days differs by less than 15% indicating that higher $O_3$ on the Top 30 days is mainly attributable to larger $O_3$ production within the modeling domain. BC contributions to the H4MDA8 are smaller than to the Top 30 in 16 of the 22 cities which is consistent with greater destruction of BC $O_3$ by local photochemistry on the highest $O_3$ days in these cities. Los Angeles and Sacramento fall in this category with BC contributions to H4MDA8 below 20 ppb. For the cities east of the Rocky Mountains the H4MDA8 contributions range from 19 ppb (St. Louis) to 34 ppb (Detroit) and the differences among the three metrics are generally within 5 ppb with no metric consistently being highest or lowest. In contrast, for cities in the Intermountain West (Boise, Phoenix, Salt Lake and Denver) the BC contribution is consistently lower for summer average than the two high $O_3$ day metrics (i.e., differences are more than 8 ppb). For Denver, our model estimates 57 ppb of BC contribution to the Top 30 and 72 ppb to the H4MDA8. However, the modeled 72 ppb BC





contribution to the Denver H4MDA8 is certainly over-stated, because the observed MDA8 on this day was only 50 ppb, and we place more emphasis on metrics like the Top 30 that consider multiple days. BC contributions tend to be higher in the western than the Eastern US because of higher terrain and deeper planetary boundary layer (PBL) that can efficiently transport mid-tropospheric $O_3$ to ground level, and longer $O_3$ lifetimes in the PBL (Fiore et al., 2002).

We further investigated how total $O_3$ changes as the modelled BC contributions increase (in 10 ppb increments) as shown in Figure 4 for several cities. The relationships vary between cities and the model captures this variation with the Western cities (Denver and Phoenix) showing different patterns than Eastern cities (Philadelphia and Atlanta). For Denver and Phoenix in the Intermountain West, total $O_3$ increases with BC contribution and approaches the 1:1 line at higher BC contribution

revealing small groups of days when MDA8 $O_3$ exceeded 60 ppb and the modelling indicates that BCs accounted for almost all of this $O_3$. These groups of days are significant because local emission reductions, or even US-wide emission reductions, would be ineffective at reducing $O_3$. However, there are other days in both Denver and Phoenix when total MDA8 $O_3$ exceeded 60 ppb with modelled BC contribution below 30 ppb (see grey circles in Figure 4) on which reducing local or US emissions would lower $O_3$. Air quality managers need methods to identify dates when emission reductions would be

ineffective so that those dates can be excluded from emission strategy development.

### 3.4 Inert vs Active Ozone BCs

Contributions to seasonal average MDA8 from reactive and inert tracers are shown in Figure 5 where positive differences indicate larger contributions from inert tracers in all seasons. In summer, when photochemistry is active, the differences are more than 10 ppb. At the Galveston and Sabine Pass sites, the differences frequently exceed 20 ppb in summer days (Figure

1). During spring and fall, contributions from inert tracers are 5 ppb higher than contributions from active tracers in Southern regions and less than 5 ppb elsewhere. In winter when photochemistry is less active the estimated contributions from the inert and active tracers are similar. Such seasonal variation is consistent across all 3 groups of vertical layers (Figure S8-S10). These results emphasize the critical role of $O_3$ chemistry and highlight the estimation bias inherent to the inert tracer approach.

### 3.5 $O_3$ Contributions by Boundary Height Range

The largest BC contributions to seasonal average MDA8 $O_3$ over the US are from the MT with small contributions from the UTLS (Figure 6). The contributions from the MT are highest in spring, followed by summer, fall and winter. The attribution of the MT to spring maxima is over 40 ppb across the Western US, but less than 25 ppb in the Eastern US. The contributions from the LT are highest (more than 30 ppb) along the western boundary but decrease sharply at the coastline as a result of

dilution as the boundary layer moves onshore along with higher $O_3$ deposition velocities over land. Dissipating contributions from the model boundaries with distance are seen at all lateral sides because $O_3$ deposits to the earth's surface and is destroyed by chemical reactions in the atmosphere. The penetration of LT BC $O_3$ inland $O_3$ is highest in winter when





chemistry and deposition are least active. The UTLS BC tracers contribute only a few ppb, mostly over the highest western terrain. The highest contributions from UTLS BCs occur in summer when vertical convection is most active.

**3.6 Sensitivity to Changing Anthropogenic Emissions (GLO and EAS Scenarios)**

Reducing emissions in East Asia by 20% (EAS scenario) decreases average $O_3$ across the US in all seasons (Figure 7, left
column). As expected, decreases are highest in the west because the Western US is closest to Asia and has high terrain. The $O_3$ decreases are small: below 1 ppb in spring and below 0.5 ppb in other seasons. Decreases in $O_3$ BC reactive tracers (Figure 7, $2^{nd}$ column) are almost identical to modeled $O_3$ decreases but slightly smaller because the reactive tracers omit some chemical interactions.

We examine more closely the relationship between changes in $O_3$ and reactive tracers in the EAS scenario. In each surface grid cell, we regress hourly $O_3$ changes against reactive tracer concentration changes (summed over boundary height ranges) to compute slope and r (Figure 7, $3^{rd}$ and $4^{th}$ columns). Slope and r values of 1 indicate that the $O_3$ changes are explained entirely by the changes in $O_3$ BC reactive tracers. The slope and r values have similar spatial patterns in all seasons. In winter and fall the slope values are near 1 with r of 0.8 to 1 across the US. In spring, strong correlation (r = 0.8 to 1) is seen in the
Western US but areas in the Eastern US have a slope lower than 0.2 and r lower than 0.4 indicating that the $O_3$ BC tracers can explain only a fraction of the total $O_3$ change, perhaps because transported $O_3$ precursors are also important in the spring. The lowest correlation (r < 0.2) is in the summertime over the Southeastern US in a region where the EAS scenario produces almost no change in surface $O_3$ indicating that transport from Asia becomes unimportant. High correlation in the Western US in all seasons emphasizes the influence of $O_3$ transport from Asia in this region.

Reducing global emissions by 20% (GLO scenario), including US emissions, decreases summertime average $O_3$ by up to 4 ppb (Figure 8). The largest reductions occur over the Eastern US where US emissions cause domestic $O_3$ production. $O_3$ reductions in fall and winter are small, generally lower than 1-2 ppb. Many NOx-rich areas (e.g., urban cores) show $O_3$ increases as a result of NOx emission reductions (i.e., NOx disbenefit) in all seasons. The spatial pattern and magnitude of
the changes in BC $O_3$ tracers differs from the changes in surface $O_3$ (Figure 8, $1^{st}$ and $2^{nd}$ columns) except near the boundaries and in winter. The correlation between changes in surface $O_3$ and BC tracers is low (|r|< 0.4) in all seasons except winter (Figure 8, $4^{th}$ column). Overall, in the GLO scenario US surface $O_3$ is more sensitive to domestic emission reductions than changes in BCs.

We use summer average MDA $O_3$ to show how the two emission scenarios change BC contributions for the 22 major cities (Figure 9). The EAS scenario reduces BC contribution in all cities with reductions range from 0.06 ppb (Houston) to 0.3 ppb (Boise). Reductions are larger in the Western US and more northern latitude in the Eastern US (e.g., larger reduction in Columbus than Atlanta). These reductions result mostly from smaller MT and LT BC contributions because the EAS



scenario scaled back the contribution of each height range about equally. The EAS scenario changed the UTLS BC contributions by less than 0.01 ppb. The GLO scenario produced larger reductions than the EAS scenario and they range from 0.4 ppb (Boston) to 1.2 ppb (Los Angeles). These reductions are mainly driven by the MT BCs in all of the cities except Dallas and Houston. Higher influence from the LT in the GLO scenario than the EAS scenario for most cities is

consistent with the GLO scenario reducing emissions just outside the CAMx domain whereas in the EAS scenario the emission reductions occur only in East Asia.

**3.7 Comparing BC O$_3$ Contributions in Two Regional Models**

The AQMEII activity permits comparison of BC O$_3$ contributions in different regional models. US EPA applied the CMAQ model over the NA domain with the same model input data as we used with CAMx except for biogenic emissions. The

WRF-CMAQ system was configured using WRFv3.4 and CMAQv5.0.2 (Appel et al., 2013; see also Foley et al., 2010 and Byun and Schere, 2006). Options in CMAQ include wet deposition as described in Byun and Schere (2006) and dry deposition as described in Pleim and Ran (2011). Additional details on the CMAQ configuration used in these simulations can be found in Solazzo et al. in this issue). Figure 10 compares BC O$_3$ contributions to seasonal average MDA8 O$_3$ estimated by CAMx and CMAQ using inert BC O$_3$ tracers (CMAQ was not run with reactive tracers). Differences between

the CMAQ and CAMx inert tracer impacts are smaller than the differences between inert and reactive tracers in CAMx (Figure 5), but they are notable, in a range of 4-8 ppb in summer and 2-6 ppb in spring with CAMx being higher. Factors contributing to these differences may include fewer vertical layers in CAMx (26, compared to 35 in CMAQ) allowing greater transport of UTLS O$_3$ to ground level (Emery et al. (2012), omission of wet scavenging for the CAMx inert tracers, treatment of deep convective transport in CMAQ, or differences in model treatments of O$_3$ dry deposition.

**4 Conclusions**

Inert BC O$_3$ tracers consistently estimate higher BC contributions to seasonal average MDA8 O$_3$ across the US than reactive tracers, particularly in summer. The inherent bias in the inert tracer approach (i.e., omitting chemical destruction) can exceed 10 ppb in seasonally averaged MDA8 O$_3$ which is substantial in comparison to the 70 ppb level of the O$_3$ NAAQS. This information is critical for interpreting results obtained with inert tracers in AQMEII-3 and other studies.

Comparing inert tracers in two regional models that used substantially the same input data found differences in MDA8 O$_3$ that were smaller than the differences between inert and reactive tracers but nevertheless were notable. Potential causes include differing numbers of model vertical layers (influencing movement of UTLS O$_3$ to ground level) and differences in model treatments of deposition.


Contributions from O$_3$ BCs in three height ranges (LT, MT and UTLS) differ spatially and temporally. The LT BC tracers do not penetrate very far inland. The largest contributions to MDA are from the MT BCs with springtime maxima exceeding 40 ppb in the high terrain of the Western US. The high contribution of BC O$_3$ to ground level O$_3$ in portions of the Western US presents a significant challenge to air quality management approaches based solely on local emission reductions.

Nonetheless, model comparison with observations suggest that estimated high BC contributions in the Intermountain West could be overstated and that the bias inherited in O$_3$ BCs can affect model performance.

Reducing emissions in East Asia (EAS scenario) revealed a near linear relationship between changes in BC O$_3$ and changes in surface O$_3$ in the Western US in all seasons and across the US in fall and winter with a near 1:1 slope. However, the

surface O$_3$ decreases are small: below 1 ppb in spring and below 0.5 ppb in other seasons. These reductions result mostly from smaller MT and LT BC contributions because the EAS scenario scaled back the contribution from each height range about equally. In the GLO scenario US surface O$_3$ is more sensitive to domestic emission reductions than changes in the BCs.

**Acknowledgement**

This study was supported by Coordinating Research Council Atmospheric Impacts Committee.

We gratefully acknowledge the contribution of various groups to the third air Quality Model Evaluation international Initiative (AQMEII) activity: U.S. EPA, Environment Canada, Mexican Secretariat of the Environment and Natural Resources (Secretaría de Medio Ambiente y Recursos Naturales-SEMARNAT) and National Institute of Ecology (Instituto

Nacional de Ecología-INE) (North American national emissions inventories); U.S. EPA (North American emissions processing and meteorology inputs); ECMWF/MACC project & Météo-France/CNRM-GAME (Chemical boundary conditions). We thank Christian Hogrefe for discussions and CMAQ results.

**Supplementary Information**

Supplementary information associated with this article can be found in the online version, at doi:xxx.

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




**Figures**

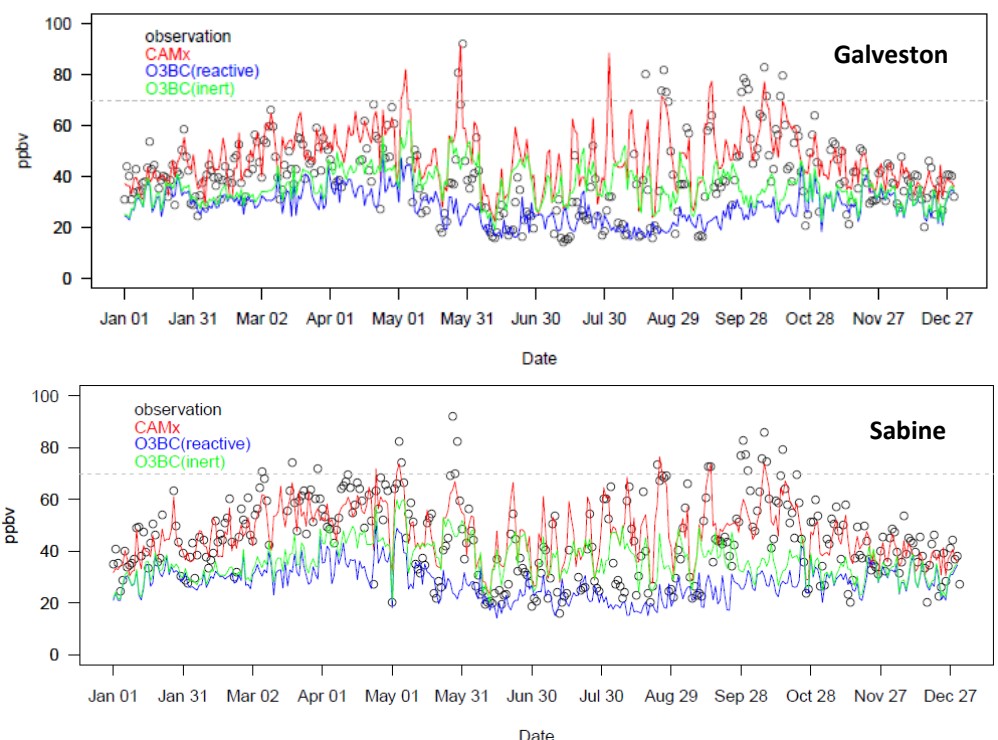

**Figure 1: Observed (black circle) and CAMx-predicted (red line) daily MDA8 $O_3$ at Galveston (top; coordinates: 29.254474°, -94.861289°), and Sabine Pass (bottom; coordinates: 29.727931°, 93.894081°) monitoring sites located on the Texas Gulf coast near Houston and Beaumont, respectively. Contributions are shown for inert (green line) and reactive (blue line) BC $O_3$ tracers summed over boundary height ranges.**




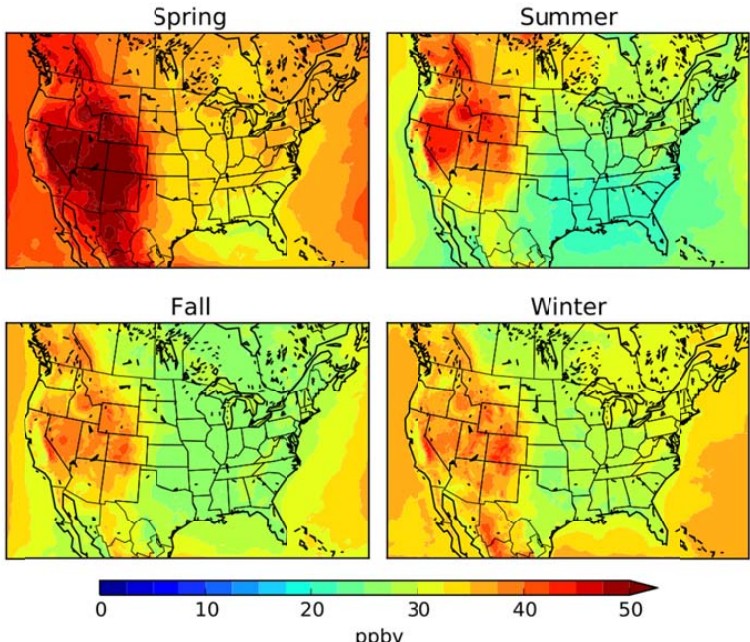

**Figure 2: BC contributions to seasonal average MDA8 O₃ from reactive tracers summed over boundary height ranges.**





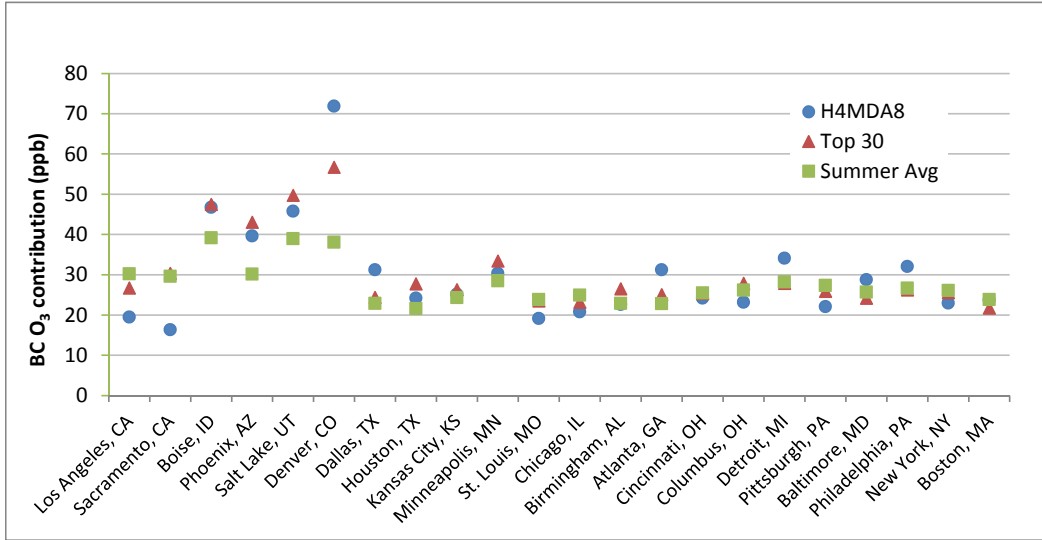

**Figure 3: BC O₃ contributions to the H4MDA (blue dot), average MDA8 on the Top 30 MDA8 O₃ days (red triangle), and summer average MDA8 (green square) at 22 major cities.**




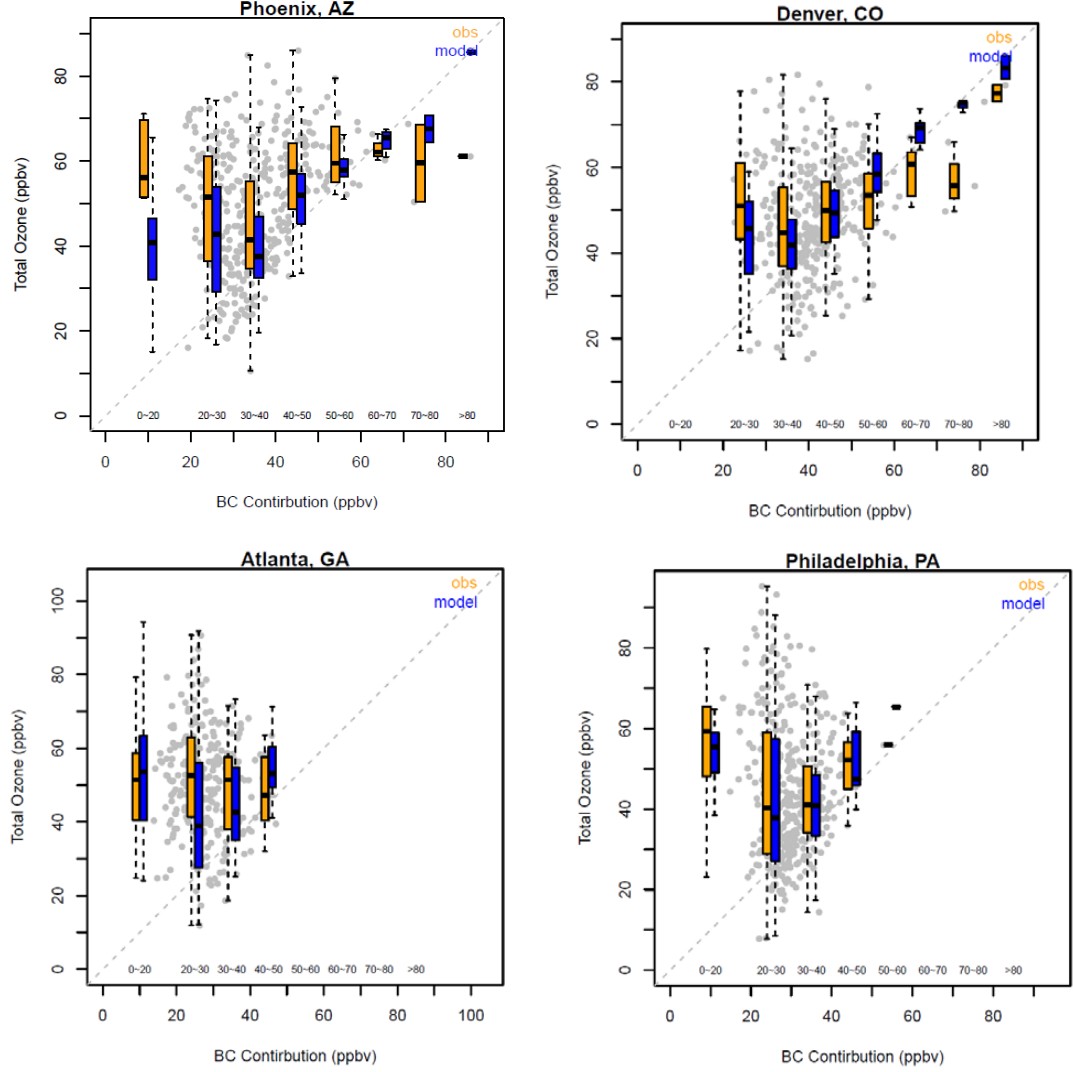

**Figure 4: Comparing total MDA8 O$_3$ to modeled BC contribution in 10 ppb ranges for observed (yellow) and modeled (blue) total O$_3$ paired in time and space shown as box-and-whisker plots. Grey circles show all observed data pairs.**





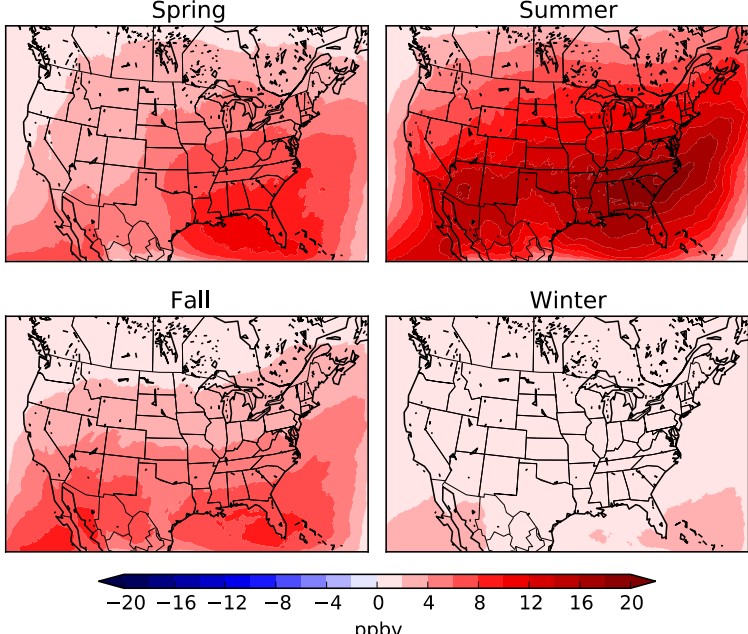

**Figure 5: Differences between inert and reactive BC O$_3$ tracer contributions to seasonally averaged MDA8 O$_3$ (inert – reactive) summed over all boundary height ranges.**





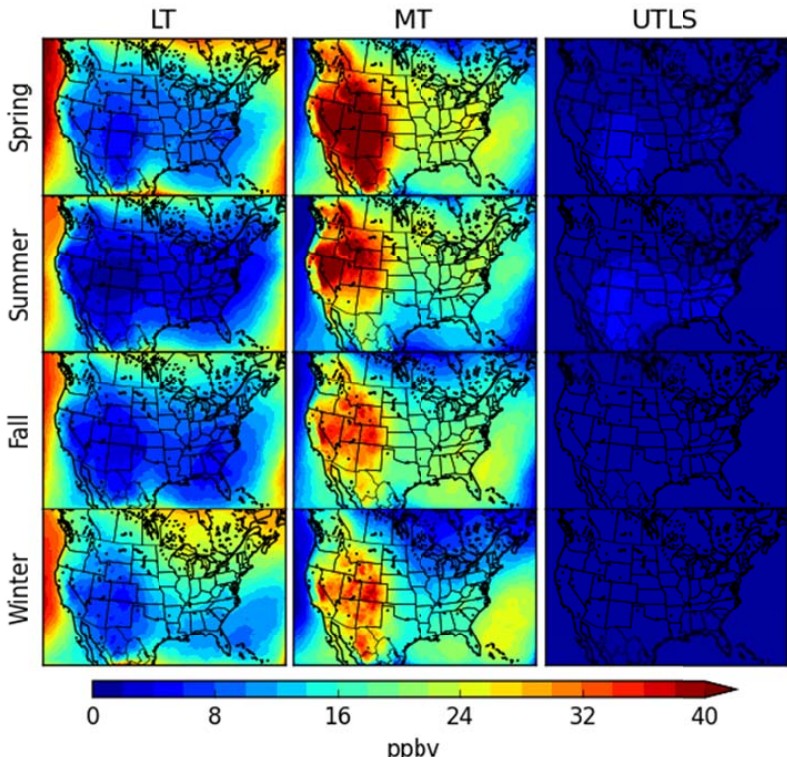

**Figure 6: Seasonal average MDA8 O$_3$ contributions from boundary height ranges using reactive BC O$_3$ tracers.**





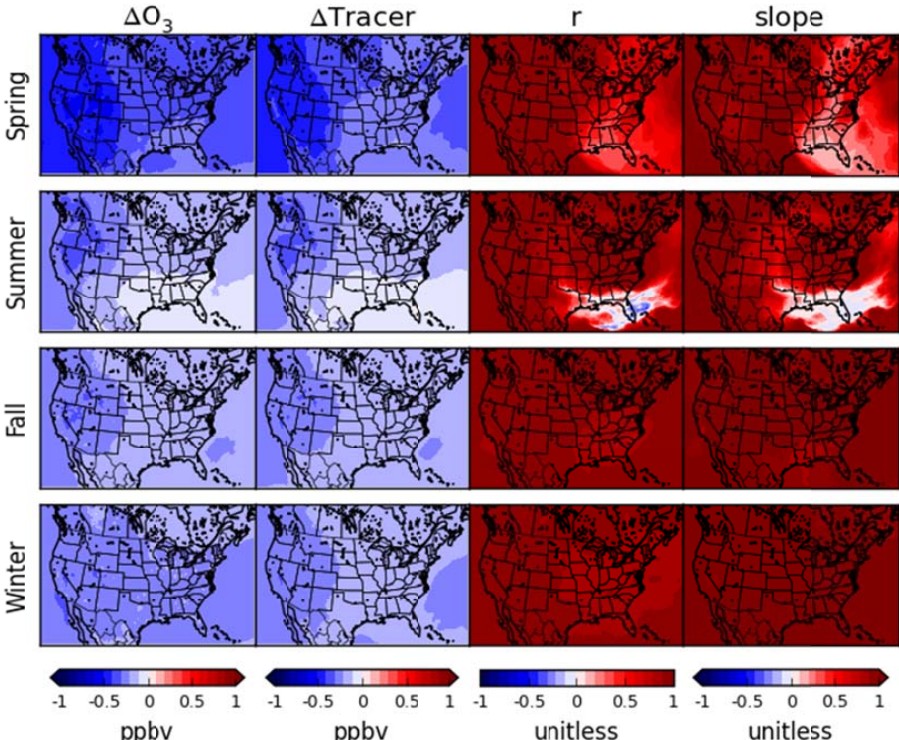

**Figure 7: Changes to seasonal average O$_3$ (left column) and reactive tracer BC O$_3$ contribution (column 2) for 20% reduction in East Asia emissions (EAS scenario). The correlation (r) and slope of a linear regression of column 2 against column 1 hourly data are shown in columns 3 and 4, respectively.**





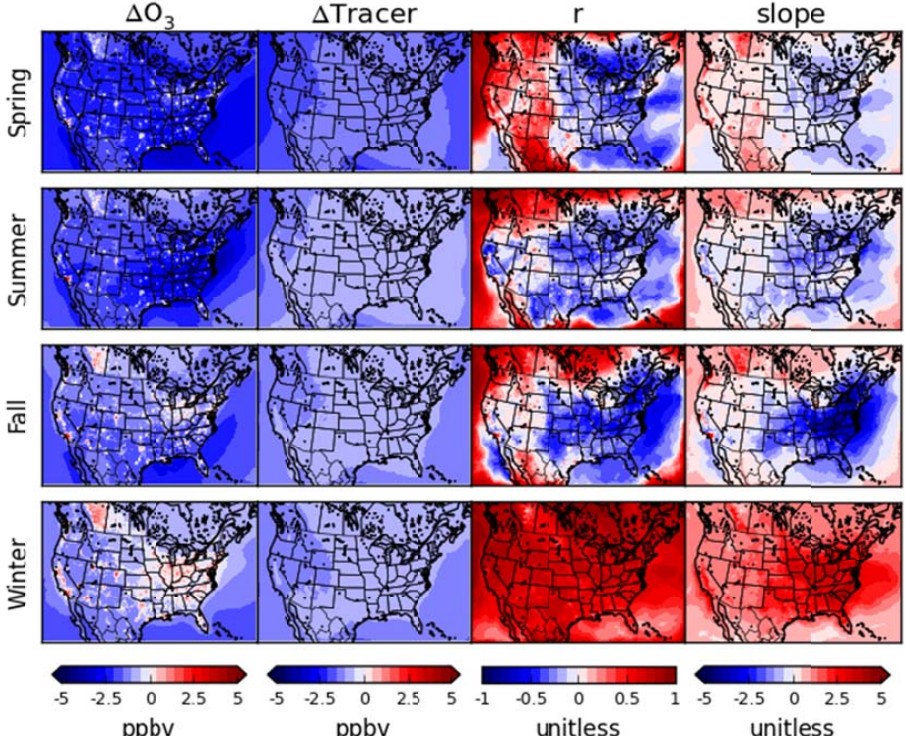

**Figure 8: Changes to seasonal average O$_3$ (left column) and reactive tracer BC O$_3$ contribution (column 2) for 20% reduction of Global emissions (GLO scenario). The correlation (r) and slope of a linear regression of column 2 against column 1 hourly data are shown in columns 3 and 4, respectively.**



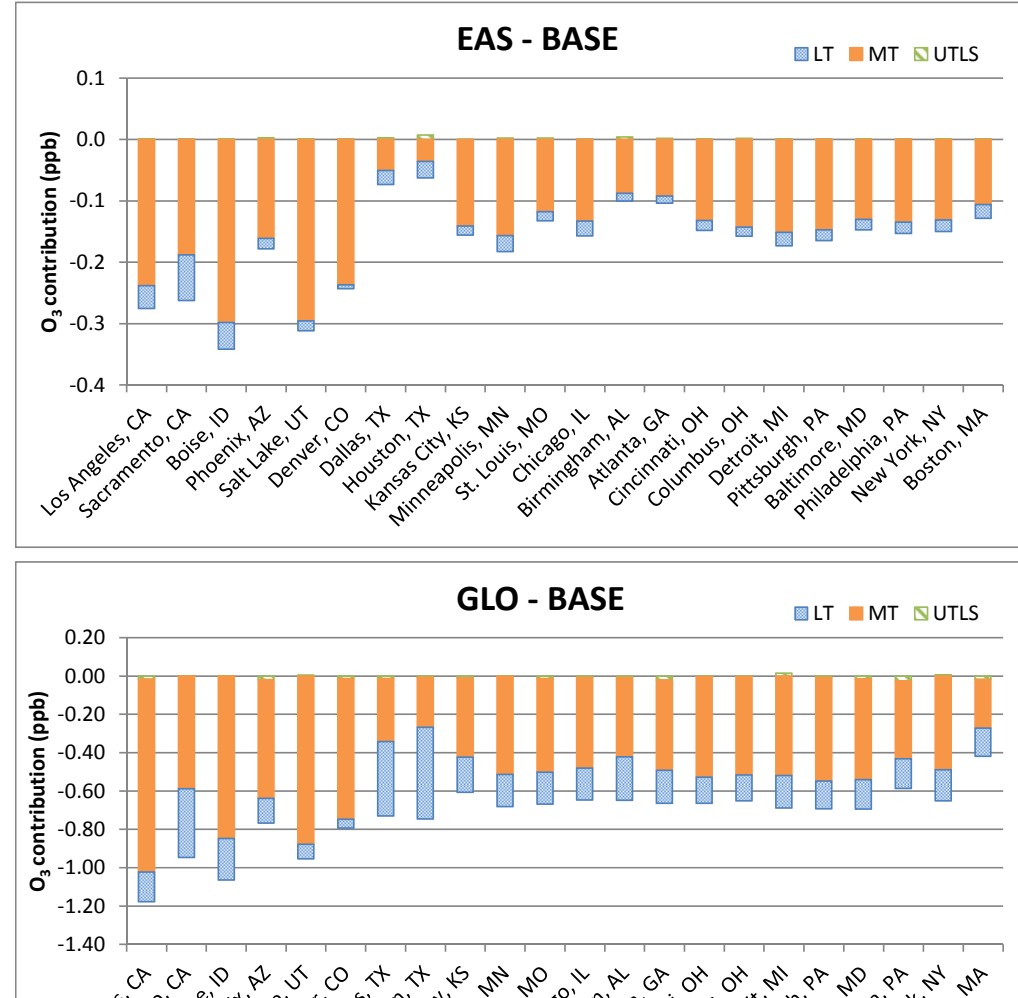

**Figure 9: Changes in BC contribution (ppb) to summer average MDA8 O₃ by height range for the EAS (top) and GLO (bottom) scenarios from the Base case.**




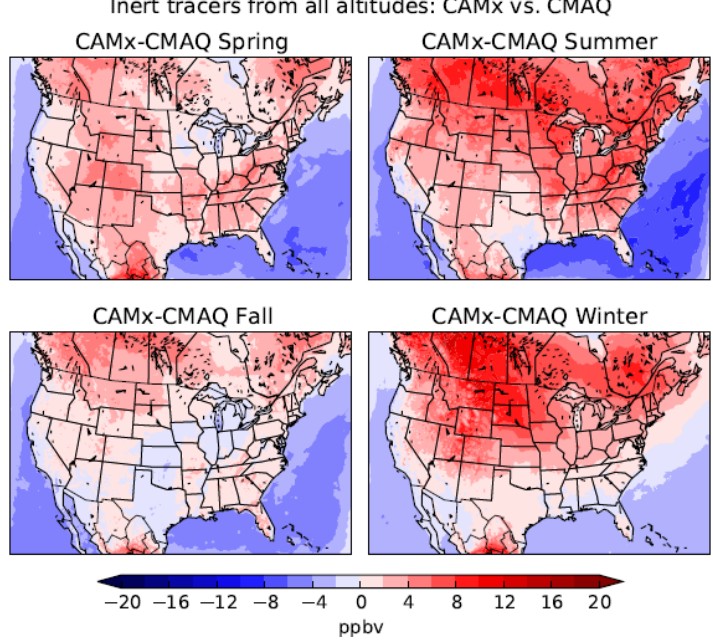

**Figure 10: Differences in seasonally averaged MDA8 contributions from inert BC O$_3$ tracers in CAMx and CMAQ (CAMx – CMAQ).**





**Tables**

**Table 1: Annual emissions by source sector in 2010 (Thousand short tons/year).**

| Sector | NOx | VOC | CO |
|---|---|---|---|
| *Elevated Sources* | | | |
| EGU (ptipm) | 2,141 | 41 | 691 |
| Non-EGU (ptnonipm) | 1,558 | 323 | 2,047 |
| International Shipping | | | |
| (c3marine) | 1,186 | 45 | 99 |
| Mexico (mexpt) | 384 | 50 | 153 |
| Canada (canpt) | 504 | 577 | 834 |
| Fire (ptfire) | 198 | 1,783 | 13,172 |
| *Low level sources* | | | |
| Anthropogenic surface | 13,067 | 15,562 | 58,672 |
| Biogenic | 728 | 53,469 | 4,309 |
| *Total* | 19,765 | 71,850 | 79,977 |

5  **Table 2: Model performance metrics where $\langle o \rangle$ is the mean and $\sigma_o$ is the standard deviation of the observed (o) or modeled (m) concentrations (C).**

| Metric (potential range) | Equation |
|---|---|
| Normalized Mean Bias (%) (-100% to +∞) Normalized Mean Error (%) (0% to +∞) | $$NMB = \frac{\sum_{i=1}^{N}(C_m - C_o)}{\sum_{i=1}^{N}C_o} \qquad NME = \frac{\sum_{i=1}^{N}\left|C_m - C_o\right|}{\sum_{i=1}^{N}C_o}$$ |
| Fractional Bias (%) (-200% to +200%) Fractional Error (%) (0% to +200%) | $$FE = \frac{1}{N}\sum_{i=1}^{N}\frac{\left|C_m - C_o\right|}{\left(\frac{C_o + C_m}{2}\right)} \quad FB = \frac{1}{N}\sum_{i=1}^{N}\frac{(C_m - C_o)}{\left(\frac{C_o + C_m}{2}\right)}$$ |
| Correlation coefficient | $$r = \frac{1}{N-1}\sum_{i=1}^{N}\left(\frac{C_m - \langle C_m \rangle}{\sigma_m}\right)\left(\frac{C_o - \langle C_o \rangle}{\sigma_o}\right)$$ |





Root Mean Square Error

$$RMSE = \sqrt{\frac{\sum_{i=1}^{N}(C_m - C_o)^2}{N}}$$

**Table 3: Seasonal and annual model performance statistics for MDA8 O$_3$ (with 40 ppb cutoff) at AQS and CASTNET sites.**

| Season | Network | #Obs | Mean | NMB (%) | NME (%) | FB (%) | FE (%) | r | RMSE |
|---|---|---|---|---|---|---|---|---|---|
| spring | AQS | 71074 | 53.6 | 2.7 | 10.9 | 2.6 | 10.8 | 0.61 | 7.44 |
| | CASTNET | 5737 | 54.2 | 1.9 | 10.3 | 1.9 | 10.2 | 0.63 | 7.11 |
| summer | AQS | 72548 | 57.3 | 5.6 | 13.4 | 5.4 | 13.0 | 0.61 | 9.53 |
| | CASTNET | 5029 | 56 | 5.5 | 12.4 | 5.6 | 12.1 | 0.63 | 8.47 |
| fall | AQS | 41729 | 48.7 | -5.9 | 12.1 | -6.4 | 12.4 | 0.68 | 8.26 |
| | CASTNET | 3597 | 47.3 | -7.3 | 10.9 | -7.6 | 11.2 | 0.71 | 7.24 |
| winter | AQS | 11823 | 40 | -10.6 | 14.0 | -11.8 | 15.1 | 0.30 | 8.35 |
| | CASTNET | 2328 | 41.3 | -9.0 | 12.1 | -10.0 | 13.1 | 0.51 | 6.67 |
| Annual | AQS | 197174 | 53.1 | 1.3 | 12.3 | 0.9 | 12.2 | 0.64 | 8.48 |
| | CASTNET | 16691 | 51.5 | -0.4 | 11.3 | -0.7 | 11.4 | 0.66 | 7.52 |

See Table 2 for definitions of the statistical metrics

