# Peer review of "Modeling inter-continental transport of ozone in North America with CAMx for the Air Quality Model Evaluation International Initiative (AQMEII) Phase 3"

_Atmospheric Chemistry and Physics, 2017_

## Referee Comment (RC1) · Anonymous Referee #1 · 11 Apr 2017

General comments:

The paper analyzes several model approaches for estimating the impact of long-range ozone transport, including use of: brute-force sensitivity tests, chemically-reactive tracers, and inert tracers. The regional CAMx modeling for the U.S. (12km) is based on well-established inputs from the AQMEII Phase 3 effort and sufficient information is presented regarding the model's ability to replicate observations from the simulation period (2010). There are several areas of focus within the manuscript: 1) the impact of boundary conditions on simulated ozone within the U.S., 2) parsing these boundary

condition impacts by height, 3) assessing the impacts of 20% reductions in emissions globally and from East Asia, 4) a limited comparison of the boundary conditions in CAMx to another regional model (CMAQ), and 5) comparisons of the various model approaches (e.g., sensitivity vs. reactive tracers) in estimating boundary impacts. The key takeaway from the manuscript is that regional models will be sensitive to biases and errors in boundary conditions, especially in the inter-mountain States in the western U.S.

The overall quality of the paper is good and the subject matter is of keen significance to the air quality management community. One general commentary on the manuscript is that there are a number of instances where a finding is made and then several hypotheses are offered for why the finding might be what it is, without any followup analyses to assess the merit of the various hypotheses. Examples include: section 3.1 ("potential causes are ..." deposition, halogen chemistry, mixing), section 3.2 ("other factors must be contributing (not examined here)"), section 3.7 ("factors contributing to these differences may include ...). Recognizing that no manuscript can be exhaustive, the authors are encouraged to reassess if more analyses are possible in the scope of this work to determine the causes for these modeled features. We especially encourage additional analyses in section 3.7 which, in its current form, raises as many questions as it answers. To the extent, that resources do not permit additional analyses, the authors are encouraged to limit the number of "dangling" hypotheses; either by saving them all for Section 4 as a sort of "next steps" list, or by deleting the sections with conclusions without identified causes.

**Specific comments:**

The two most likely "policy-relevant" conclusions to be cited from this manuscript are that 1) boundary conditions impacts on the Denver area average 57 ppb on the days with the highest MDA8 O3, and 2) that a 20% reduction in emissions from East Asia will have < 1 ppb impact on surface O3 in the U.S. Particularly for that first conclusion, the manuscript would be improved if more detail was provided about the robustness

of the conclusion. For instance, it is not clear to this reader whether the city-specific analyses are based on a single site or an aggregate of sites within an area. Additionally, given the note in the paper about the high modeled bias on the the H4MDA8 O3 day (observed O3 = 50 ppb while BC impacts alone > 70 ppb), it would be helpful if the model bias/error values for the top 30 subset of days were also included in table 3 or elsewhere. Given the paper's conclusion that biases/errors in the boundary conditions will affect regional concentrations, it is imperative to understand what the biases/errors are on these Top 30 days before too much weight is assigned to the 57 ppb conclusion in Denver (i.e., if there's a positive bias in O3 over those 30 days, then that specific estimate of the role of BC may also be overestimated). The Denver area is notoriously hard to model. Are the authors comfortable that the 12km CAMx modeling is properly capturing the meteorology ("Denver cyclone") and other daily-varying conditions that lead to a complex mix of local/regional/natural/international contributions in this area? It's hard to discern that from seasonal-average tables of bias and error.

Per the finding that there is a near-linear relationship between the O3 changes in the boundary conditions and the surface O3 changes in the western U.S., might there be a more direct way to visualize this finding than the 16-panel plots? Seems like scatterplots of delta O3 vs. delta tracer would show this conclusion more directly (by region, if needed). Alternatively, perhaps spatial maps of percent O3 or tracer changes (as opposed to absolute change) would make the point more directly.

Per Figure 4, can the plot be modified to show the count of data points in each box/whisker. If there are some boxes with less than some small number of data, perhaps those should be combined into a larger range w/ more statistical robustness. Would it be possible for the authors to comment on an additional possible conclusion from Denver/Fig 4? It appears to me that the model is overestimating the BC -> total O3 slope in this area (in Phoenix as well). The BC/total slope in the model appears to be close to 1 (i.e., what distinguishes high days from low days in Denver is BC contributions), whereas the observations suggest something much flatter (i.e., what

СЗ

distinguishes high days from low days in Denver is something other than BC). This seems like a potentially important finding. Once the model exceeds 50 ppb, the BC terms are large (and appear to be overdone).

---

## Referee Comment (RC2) · Anonymous Referee #2 · 12 Apr 2017

The work presented in an interesting contribution to the scope of the ACP Special Issue and fits very well within the papers that I have seen published so far.

In fact in its relative simplicity it addresses questions that are nicely linking the global and the regional scale in two relevant ways: the influence of long range transport on regional scale chemical budgets (read Ozone); but also the possible influences of global scale models on regional scale ones, which acquires from the first the mass budget at the boundaries.

The paper is well written. If I have to find a criticism I would say that it is probably

too well written or better to little written. What I mean by that is that in many places, basically all sections the explanations as well as the writing style are a bit too concise and to the essence. A contrast is present between the will to present facts and figures and the fact that most of the time qualitative definitions like, "large, bigger, better, acceptable" are use to characterize the results.

The figures are nice and give clear quantitative indication of the various aspects that the study tackles, which however is not reflected in the text at times. More elaboration and quantification is needed here and there to make the story more interesting and appealing and to elevate the valuable content of this paper from a report style.

I am not asking to re-write the paper here, cause that would be unfair, just to indulge in a deeper explanation of the results by deepening only into those explanations that are worth exploring.

Cities are taken as locations for the comparison with data. It is important to characterize the monitoring sites and clarify whether those location are suitable to measure background levels of ozone. The inert ozone tracer is a powerful tool that should be exploited more in the future, considering the relevance of the impact of BC bias on regional scale models. A more detail break down in the vertical would be very instructive when studying for example boundary layer exchanges or transition from marine to land ABL. But this is probably for the future.

---

## Author Comment (AC1) · 23 Jun 2017

Responses to Reviewer Comments

**Modeling inter-continental transport of ozone in North America with CAMx for the Air Quality Model Evaluation International Initiative (AQMEII) Phase 3**

doi:10.5194/acp-2017-194, 2017.

Uarporn Nopmongcol et al.

The authors thank the reviewer for their helpful comments.  Below we respond to each and note our changes to the manuscript.

**Anonymous Referee #1

General comments: The paper analyzes several model approaches for estimating the impact of long-range ozone transport, including use of: brute-force sensitivity tests, chemically-reactive tracers, and inert tracers. The regional CAMx modeling for the U.S. (12km) is based on well-established inputs from the AQMEII Phase 3 effort and sufficient information is presented regarding the model's ability to replicate observations from the simulation period (2010). There are several areas of focus within the manuscript: 1) the impact of boundary conditions on simulated ozone within the U.S., 2) parsing these boundary condition impacts by height, 3) assessing the impacts of 20% reductions in emissions globally and from East Asia, 4) a limited comparison of the boundary conditions in CAMx to another regional model (CMAQ), and 5) comparisons of the various model approaches (e.g., sensitivity vs. reactive tracers) in estimating boundary impacts. The key takeaway from the manuscript is that regional models will be sensitive to biases and errors in boundary conditions, especially in the inter-mountain States in the western U.S. The overall quality of the paper is good and the subject matter is of keen significance to the air quality management community. One general commentary on the manuscript is that there are a number of instances where a finding is made and then several hypotheses are offered for why the finding might be what it is, without any followup analyses to assess the merit of the various hypotheses. Examples include: section 3.1 ("potential causes are ..." deposition, halogen chemistry, mixing), section 3.2 ("other factors must be contributing (not examined here)"), section 3.7 ("factors contributing to these differences may include ...). Recognizing that no manuscript can be exhaustive, the authors are encouraged to reassess if more analyses are possible in the scope of this work to determine the causes for these modeled features. We especially encourage additional analyses in section 3.7 which, in its current form, raises as many questions as it answers. To the extent, that resources do not permit additional analyses, the authors are encouraged to limit the number of "dangling" hypotheses; either by saving them all for Section 4 as a sort of "next steps" list, or by deleting the sections with conclusions without identified causes.

Response 1: We agree that further investigation into difference between CAMx and CMAQ (and other model) results will be useful and note that US EPA is leading this effort and their paper will be submitted to this same Special Issue. We have added their paper to our reference list (Liu et al.).  The point of Section 3.7 is to emphasize the magnitude of bias using the inert tracer approach relative to chemically-reactive tracers, and the fact that this bias is much larger than differences between two inert models using the same inputs. We have added the following text in bold to Section 3.7.

"Differences between the CMAQ and CAMx inert tracer impacts are smaller than the differences between inert and reactive tracers in CAMx which exceed 10 ppb (Figure 5), but they are notable, in a range of 4-8 ppb in summer and 2-6 ppb in spring with CAMx being higher. Factors contributing to these differences may include fewer vertical layers in CAMx (26 compared to 35 in CMAQ), which may cause

more numerical diffusion of UTLS $O_3$ to ground level (Emery et al. (2012), omission of wet scavenging for the CAMx inert tracers, treatment of deep convective transport in CMAQ, or differences in model treatments of $O_3$ dry deposition. **Liu et al. (this issue) performed multi-model process comparisons with four AQMEII models and draw similar conclusions regarding factors that can contribute to differences in tracer impacts.**"

Reference: Liu, P., Hogrefe, C., Bieser, J., Im, U., Mathur, R., Nopmongcol., U., Roselle, S., Spero, T.: Multi-Model Comparison of Lateral Boundary Contributions to Surface Ozone Over the United States, in preparation.

We think including our hypotheses within their respective sections (the way the text is currently written) is useful. As suggested by the reviewer, we have also summarized these hypotheses in Section 4 as suggestions for future work.

Specific comments: The two most likely "policy-relevant" conclusions to be cited from this manuscript are that 1) boundary conditions impacts on the Denver area average 57 ppb on the days with the highest MDA8 O3, and 2) that a 20% reduction in emissions from East Asia will have < 1 ppb impact on surface O3 in the U.S. Particularly for that first conclusion, the manuscript would be improved if more detail was provided about the robustness of the conclusion. For instance, it is not clear to this reader whether the city-specific analyses are based on a single site or an aggregate of sites within an area. Additionally, given the note in the paper about the high modeled bias on the the H4MDA8 O3 day (observed O3 = 50 ppb while BC impacts alone > 70 ppb), it would be helpful if the model bias/error values for the top 30 subset of days were also included in table 3 or elsewhere. Given the paper's conclusion that biases/errors in the boundary conditions will affect regional concentrations, it is imperative to understand what the biases/errors are on these Top 30 days before too much weight is assigned to the 57 ppb conclusion in Denver (i.e., if there's a positive bias in O3 over those 30 days, then that specific estimate of the role of BC may also be overestimated). The Denver area is notoriously hard to model. Are the authors comfortable that the 12km CAMx modeling is properly capturing the meteorology ("Denver cyclone") and other daily-varying conditions that lead to a complex mix of local/regional/natural/international contributions in this area? It's hard to discern that from seasonal-average tables of bias and error.

Response 2: We thank the reviewer for this suggestion. We have added quantile-quantile (Q-Q) plots to the SI (Figure S8; Denver example is shown in Figure R1). Q-Q plots compare the independently sorted (time-unpaired) values of observed and modeled concentrations and are useful for evaluating whether the model can reproduce the distribution of observed values (perfect model performance would show all data points along a 1:1 line). Our Q-Q plots suggest good model distributions (e.g., data points nearly match the 1:1 line) for Los Angeles, Sacramento, Phoenix, Denver, Dallas, Houston, Pittsburgh, and Philadelphia. In Denver, the model captures the ozone distribution below 65 ppb well and slightly under predicts ozone above 65 ppb, but it over predicts the highest MDA8 ozone. We have added the following statement (text in bold) to the second paragraph of Section 3.1.

[Figure]

Figure R1. Example of Q-Q plot for Denver

**"We additionally provide quantile-quantile (Q-Q) plots (Figure S8) which compare independently sorted (time-unpaired; space-paired) observed and modeled $O_3$ for each city. The Q-Q plots suggest good model distributions (e.g., data pairs near the 1:1 line) for Los Angeles, Sacramento, Phoenix, Denver, Dallas, Houston, Pittsburgh, and Philadelphia."**

The city analyses are based on a single site that shows the highest H4MDA8 within the metropolitan statistical area as described at the end of Section 2.2 and in Figure S1. The selected AQS site ID is provided in Table S3.

Per the finding that there is a near-linear relationship between the O3 changes in the boundary conditions and the surface O3 changes in the western U.S., might there be a more direct way to visualize this finding than the 16-panel plots? Seems like scatterplots of delta O3 vs. delta tracer would show this conclusion more directly (by region, if needed). Alternatively, perhaps spatial maps of percent O3 or tracer changes (as opposed to absolute change) would make the point more directly.

Response 3:  We added two scatter plots to Figure 7 (See Figure R2 below) to help explain the information offered in this figure. The two scatter plots show the relationship of delta O3 vs. delta tracer for Denver in spring and summer. The scater plots also illustrate the regression analyses that we performed to develop Figure 7. Columns 3 and 4 in Figure 7 show the regression parameters for each surface grid cell and match the scatter plots for Denver. We have revised our text (shown in bold) in the second paragraph of Section 3.6 to better describe this plot.

"We examine more closely the relationship between changes in $O_3$ and reactive tracers in the EAS scenario. In each surface grid cell, we regress hourly $O_3$ changes against reactive tracer concentration changes (summed over boundary height ranges) to compute slope and r **as demonstrated in the two scatter plots for Denver in spring and summer**. Slope and r values of 1 indicate that the $O_3$ changes are explained entirely by the changes in $O_3$ BC reactive tracers. **The delta total $O_3$ and delta tracer $O_3$ relationship is near-linear at Denver with a slope of 0.87 and r of 0.9. The 16 panels in Figure 7 show the regression parameters for each grid surface cell and match the scatter plots for Denver.** The slope and r (Figure 7, 3rd and 4th columns) values have similar spatial patterns in all seasons. In winter and fall the slope values are near 1 with r of 0.8 to 1 across the US **suggesting strong influence of $O_3$ transport from Asia during these seasons**. In spring, strong correlation (r = 0.8 to 1) is seen in the Western US but areas in the Eastern US have a slope lower than 0.2 and r lower than 0.4 indicating that the $O_3$ BC tracers

can explain only a fraction of the total O$_3$ change. The lowest correlation (r < 0.2) is in the summertime over the Southeastern US in a region where the EAS scenario produces almost no change in surface O$_3$ indicating that transport from Asia becomes unimportant. High correlation in the Western US in all seasons emphasizes the influence of O$_3$ transport from Asia in this region."

[Figure]

**Figure R2: Scatter plots on the top show delta daily average O$_3$ (y-axis) and reactive tracer BC O$_3$ contribution (x-axis) for Denver in spring and summer for 20% reduction in East Asia emissions (EAS scenario). The 16 panels summarize the same information showing seasonal delta total O$_3$ (left column) and reactive tracer BC O$_3$ contribution (column 2) for each grid surface. The correlation (r) and slope of a linear regression of column 2 against column are shown in columns 3 and 4, respectively.**

Per Figure 4, can the plot be modified to show the count of data points in each box/whisker. If there are some boxes with less than some small number of data, perhaps those should be combined into a larger range w/ more statistical robustness.

Response 4: We have included the count of predicted data points in Figure 4 as suggested by the reviewer. We prefer not to group data into larger ranges because it creates uneven spacing and loses detail. Additional modification to Figure 4 is described in our Response#5 below.

Would it be possible for the authors to comment on an additional possible conclusion from Denver/Fig 4? It appears to me that the model is overestimating the BC -> total O3 slope in this area (in Phoenix as well). The BC/total slope in the model appears to be close to 1 (i.e., what distinguishes high days from low days in Denver is BC contributions), whereas the observations suggest something much flatter (i.e., what distinguishes high days from low days in Denver is something other than BC). This seems like a potentially important finding. Once the model exceeds 50 ppb, the BC terms are large (and appear to be overdone).

Response 5: We agree that the model tends to overestimate BC on the highest days and this is the reason we look at multiple ozone metrics including summer average and Top 30 days. The model is not perfect, but it captures the ozone distribution at Denver quite well based on the Q-Q plot we added to the SI (Figure S8 in the SI; Figure R3 in this document). The conclusions we can draw from Figure 4 are limited because BC contributions cannot be derived from measurements. We have decided that including observations in Figure 4 is confusing and taking away the key point we want to make (i.e., different pattern of how total $O_3$ changes as the modelled BC contributions increase seen in the west and the east). So we removed observations (yellow bars) from this plot and revised the relevant text in Section 3.3 as shown in bold below.

[Figure]

**Figure R3: Box-and-whisker plots comparing modeled total MDA8 O$_3$ to modeled BC contribution in 10 ppb ranges paired in time and space.**

"We further investigated how total O$_3$ changes as the modelled BC contributions increase (in 10 ppb increments) as shown in Figure 4 for several cities. The relationships vary between cities and the model captures this variation with the Western cities (Denver and Phoenix) showing different patterns than Eastern cities (Philadelphia and Atlanta).  For Denver and Phoenix in the Intermountain West, total O$_3$ increases with BC contribution and approaches the 1:1 line at higher BC contribution revealing small groups of days when MDA8 O$_3$ exceeded 60 ppb **(11 days for Denver and 7 days for Phoenix)** and the modelling indicates that BCs accounted for almost all of this O$_3$. **In other words, BC contributions alone distinguish high O3 days from low days.** These groups of **high BC-contributed** days are important because local emission reductions, or even US-wide emission reductions, would be ineffective at reducing O$_3$. However, there are other days in Denver when total MDA8 O$_3$ exceeded 60 ppb with modelled BC contribution below 30 ppb (see first and second bars in Figure 4) on which reducing local or US emissions would lower O$_3$. Air quality managers need methods to identify dates when emission reductions would be ineffective so that those dates can be excluded from emission strategy development. **Nonetheless, these results should be interpreted with consideration given to model performance. As shown in the Q-Q plot for Denver (Figure S8), the model can capture the O$_3$ distribution quite well, although it underestimates MDA8 O$_3$ over 65 ppb and overestimates MDA8 O$_3$ over 80 ppb. For this reason, we encourage making use of multi-day metrics (such as Top 30) rather than a single-day metric (e.g., H4MDA8)** "

---

## Author Comment (AC2) · 23 Jun 2017

Responses to Reviewer Comments
**Modeling inter-continental transport of ozone in North America with CAMx for the Air Quality Model Evaluation International Initiative (AQMEII) Phase 3**

doi:10.5194/acp-2017-194, 2017.

Uarporn Nopmongcol et al.

The authors thank the reviewer for their helpful comments.  Below we respond to each and note our changes to the manuscript.

**Anonymous Referee #2

The work presented in an interesting contribution to the scope of the ACP Special Issue and fits very well within the papers that I have seen published so far. In fact in its relative simplicity it addresses questions that are nicely linking the global and the regional scale in two relevant ways: the influence of long range transport on regional scale chemical budgets (read Ozone); but also the possible influences of global scale models on regional scale ones, which acquires from the first the mass budget at the boundaries. The paper is well written. If I have to find a criticism I would say that it is probably too well written or better to little written. What I mean by that is that in many places, basically all sections the explanations as well as the writing style are a bit too concise and to the essence. A contrast is present between the will to present facts and figures and the fact that most of the time qualitative definitions like, "large, bigger, better, acceptable" are use to characterize the results.

Response 1:  We have checked to make sure that any qualitative characterization is always accompanied by numerical statement. For example, "CAMx has a large over prediction bias exceeding 15 ppb at the two sites when observations are low (~20 ppb)."

The figures are nice and give clear quantitative indication of the various aspects that the study tackles, which however is not reflected in the text at times. More elaboration and quantification is needed here and there to make the story more interesting and appealing and to elevate the valuable content of this paper from a report style. I am not asking to re-write the paper here, cause that would be unfair, just to indulge in a deeper explanation of the results by deepening only into those explanations that are worth exploring.

Response 2:  We thank the reviewer for this suggestion. We have added more analyses and expanded our conclusion section to better capture essence of our story.  Please see our modified conclusion as shown in bold text below.

"
**The overall MDA8 $O_3$ performance is within evaluation goals. We do not see evidence of systematic problems with the model setup, although performance at individual monitor does vary, and the potential for hidden biases and errors always exists. Future studies could benefit from refining model assumptions that may be important at specific sites. For example, overstated MBL $O_3$ at Trinidad Head is partly attributable to the lack of $O_3$ destruction by oceanic halogen chemistry. Other possible reasons include insufficient $O_3$ deposition to the ocean, too strong vertical mixing in the MBL, or a combination of factors. The model tendency to overestimate $O_3$ in spring may suggest overstated BC contributions as seen at Denver site. Perfecting model performance at individual sites across the US is not pursued in the current study. If accuracy in estimating BC contributions is critical, such as in demonstrating attainment of $O_3$ standards, model performance and BC contributions especially on high $O_3$ days cannot be overlooked.**

Inert BC $O_3$ tracers consistently estimate higher BC contributions to seasonal average MDA8 $O_3$ across the US than reactive tracers, particularly in summer. The inherent bias in the inert tracer approach (i.e., omitting chemical destruction) can exceed 10 ppb in seasonally averaged MDA8 $O_3$ which is substantial in comparison to the 70 ppb level of the $O_3$ NAAQS. This information is critical for interpreting results obtained with inert tracers in AQMEII-3 and other studies.

Comparing inert tracers in two regional models that used substantially the same input data found differences in MDA8 $O_3$ that were **generally within 5 ppb**, smaller than the differences between inert and reactive tracers **run in a single model (CAMx),** but nevertheless **those inert model differences** were notable. Potential causes include differing numbers of model vertical layers (influencing movement of UTLS $O_3$ to ground level) and differences in model treatments of deposition. **This exercise emphasizes that source contribution analyses of BC $O_3$ (or other non-inert pollutants) using the inert tracer approach should only be interpreted qualitatively especially during the spring and summer period. Making tracers reactive is a simple improvement that is very important to this type of analysis. Future studies should consider adopting the reactive tracer approach.**

Contributions from $O_3$ BCs in three height ranges (LT, MT and UTLS) differ spatially and temporally. The LT BC tracers do not penetrate very far inland **with contributions to MDA8 $O_3$ up to 20 ppb in the coastal states**. The largest contributions to MDA8 $O_3$ are from the MT BCs with springtime maxima exceeding 40 ppb in the high terrain of the Western US. The high contribution of BC $O_3$ to ground level $O_3$ in portions of the Western US presents a significant challenge to air quality management approaches based solely on local emission reductions. Nonetheless, model comparison with observations suggest**s** that estimated high BC contributions in the Intermountain West could be overstated and that the bias inherent in $O_3$ BCs can affect model performance. **Replicating the highest end of observed $O_3$ distribution is particularly challenging. We encourage adopting multi-day metrics (such as Top 30) as an alternative to a single-day metric (e.g., H4MDA8) when examining contributions from international transport.**

Reducing emissions in East Asia (EAS scenario) revealed a near linear relationship between changes in BC $O_3$ and changes in surface $O_3$ in the Western US in all seasons and across the US in fall and winter with a near 1:1 slope. However, the surface $O_3$ decreases are small: below 1 ppb in spring and below 0.5 ppb in other seasons. These **$O_3$** reductions result mostly from reductions in MT and LT BC contributions. **Our 2010 EAS contribution results are slightly higher than the estimates of 0.35-0.45 ppb from multi-model experiment that also simulated the EAS scenario but for the year 2001 (Reidmiller et al.,2009). This is expected as East Asia emissions have increased over the last decade. Assuming a linear relationship, our study suggests an EAS total contribution of 2.5 ppb in certain seasons based on 0.5 ppb $O_3$ reduction with 20% emission decrease. It is difficult to quantify how the model biases affect the $O_3$ response to emission perturbations because the sources of biases are unknown.** In the GLO scenario US surface $O_3$ is more sensitive to domestic emission reductions than changes in the BCs.

"

Cities are taken as locations for the comparison with data. It is important to characterize the monitoring sites and clarify whether those location are suitable to measure background levels of ozone.

Response 3: Our interest is high ozone and the selected locations have high ozone exposure. For this reason we look at the AQS rather than CASTNET monitors. The 22 major cities are selected based on their population size (~200K in Salt Lake to 8.5M in New York) and geographical location. If a city has multiple AQS monitors, then we select the one with highest H4MDA8. This is described at the end of Section 2.2 and in Figure S1. The selected AQS site ID is provided in Table S3.

The inert ozone tracer is a powerful tool that should be exploited more in the future, considering the relevance of the impact of BC bias on regional scale models.

Response 4:  In our view, inert tracers are useful, but subject to a clear bias. Making tracers reactive is a simple improvement that is very important to this type of analysis and the community should adopt that approach.

A more detail break down in the vertical would be very instructive when studying for example boundary layer exchanges or transition from marine to land ABL. But this is probably for the future.

Response 5:  We agree with this future-work recommendation.